# Peptide signaling without feedback in signal production operates as a true quorum sensing communication system in *Bacillus subtilis*

Iztok Dogsa [1,3 ✉], Mihael Spacapan[1], Anna Dragoš [1,2], Tjaša Danevčič[1], Žiga Pandur[1] & Ines Mandic-Mulec [1,3 ✉]

Bacterial quorum sensing (QS) is based on signal molecules (SM), which increase in concentration with cell density. At critical SM concentration, a variety of adaptive genes sharply change their expression from basic level to maximum level. In general, this sharp transition, a hallmark of true QS, requires an SM dependent positive feedback loop, where SM enhances its own production. Some communication systems, like the peptide SM-based ComQXPA communication system of *Bacillus subtilis*, do not have this feedback loop and we do not understand how and if the sharp transition in gene expression is achieved. Based on experiments and mathematical modeling, we observed that the SM peptide ComX encodes the information about cell density, specific cell growth rate, and even oxygen concentration, which ensure power-law increase in SM production. This enables together with the cooperative response to SM (ComX) a sharp transition in gene expression level and this without the SM dependent feedback loop. Due to its ultra-sensitive nature, the ComQXPA can operate at SM concentrations that are 100–1000 times lower than typically found in other QS systems, thereby substantially reducing the total metabolic cost of otherwise expensive ComX peptide.

[1] Chair of Microbiology, Department of Food Science and Technology, Biotechnical Faculty, University of Ljubljana, Večna pot 111,1000, Ljubljana, EU, Slovenia. [2] Department of Biotechnology and Biomedicine, Section for Microbial and Chemical Ecology, Bacterial Interactions and Evolution, Technical University of Denmark, Søltofts Plads Building: 221, 164, 2800 Kgs., Lyngby, EU, Denmark. [3]These authors contributed equally: Iztok Dogsa, Ines Mandic-Mulec. ✉email: iztok.dogsa@bf.uni-lj.si; ines.mandicmulec@bf.uni-lj.si

Bacteria secrete and share quorum-sensing (QS) signal molecules (SM) that bind to specific receptors and induce cell density-dependent adaptive responses[1,2] and affect microbial community interactions when the critical concentration of SM is reached[3,4]. Not every bacterial communication system is QS (Fig. 1a)—only the communication systems where bacterial response follows a sharp transition dynamics from basic to the maximum response level is true quorum sensing[5]. To achieve this, most QS systems incorporate the coupling between signal production and signal detection with the signal amplifying its own product, although maximum response level can be reached also when signal auto-amplification is artificially broken[6]. In contrast, the dynamics of signal molecule synthesis and response in communication[7,8] systems without positive feedback loop regulation has been little studied so far, especially, quantitative research on encoder (signal molecule production) and decoder modules (response to signal molecule) comprising communication system is missing[9]. We close this knowledge gap by studying the ComQXPA communication system of *Bacillus subtilis*, where signal molecule production is not coupled to signal molecule detection[10–13].

In general, quorum sensing encoder module that encodes the information about the cell density into the signal molecule (SM) concentration can be classified according to its sensitivity to cell density. When SM concentration increases faster than linearly with cell concentration, one can call such encoder module ultra-sensitive[9]. The classification of decoder modules seems to be less complete, therefore, we follow the general definition of ultra-sensitivity in molecular biology: ultra-sensitivity describes an output response that is more sensitive to stimulus change than the hyperbolic Michaelis–Menten response[14]. In the communication systems the stimulus is signal molecule (SM) and response (RM) is the expression level of gene dependent on SM. The ultra-sensitive communication system has ultra-sensitive encoder and decoder module.

The ComQXPA communication system is likely not unique to *B. subtilis* species and its close relatives, as *comQXPA*-like loci are predicted to occur outside the *B. subtilis* clade, including some species of *Clostridiales* order[15]. No known regulators of *comQXPA* operon expression exist, it is however known that the operon is not expressed in the presence of superoxide radicals[16]. The ComQXPA communication system (Fig. 1b) involves the ComQ isoprenyl-transferase, which is required for the maturation of the ComX signal peptide. The mature ComX, which is a signal molecule (SM) of interest in this manuscript, is first synthesized as a 55-residue propeptide and then processed and modified by ComQ[11,17]. Depending on the strain-specific type (pherotype)[18], the mature ComX exists as isoprenylated 5–10 amino-acid peptide[19] that once secreted can bind to the membrane receptor histidine kinase ComP, which triggers phospho-transfer to ComA[17,18,20]. Phosphorylated ComA directly modulates the expression level of various genes, including the expression level of *srfA* operon[12], which serves as a measure for the response (RM) to the signal molecule SM (Fig. 1b). Although the *srfA* expression required for the synthesis of the major lipopeptide antibiotic surfactin[20] also depends on other extracellular peptide signaling systems from the Rap-Phr family[21,22], the research of this paper is focused on ComX dependent response. For true quorum sensing, which is regarded as a population density-driven event, one would expect that most of the cells will be involved in the communication. Ideally, every cell produces signaling molecules, shares signaling molecules, and coordinately responds to the signaling molecules. In recent years, however, it was shown that the expression of signaling molecules can be heterogenous[23–26]. When the concentration of signaling molecule reaches a threshold value, a coordinated and homogeneous expression of target genes may be initiated in all cells of the population[5,27] or a heterogeneous gene expression in the population may be triggered at low concentrations[25,27,28]. However, these studies were performed on communication systems with positive feedback loop regulation, where signal molecule (SM) enhances its own production. How heterogeneous is the population of signal producers and responders in communication systems lacking feedback loop regulation, like the ComQXPA in *B. subtilis*, is to the best of our knowledge, unknown.

It was theoretically estimated that among SM, peptide signals of Gram-positive bacteria are more than 20 times metabolically more expensive than AHLs produced by Gram-negative bacteria[29]. The existence of fitness cost of signal molecule production in Gram-negative bacterial models has been theoretically predicted[30–32] and experimentally supported[33]. One would thus expect that the fitness burden of metabolically costly SM production in Gram-positives is even more pronounced.

In order to determine the operational mode of the communication system without the SM dependent feedback loop, we quantified the system's core parameters dynamics by modeling

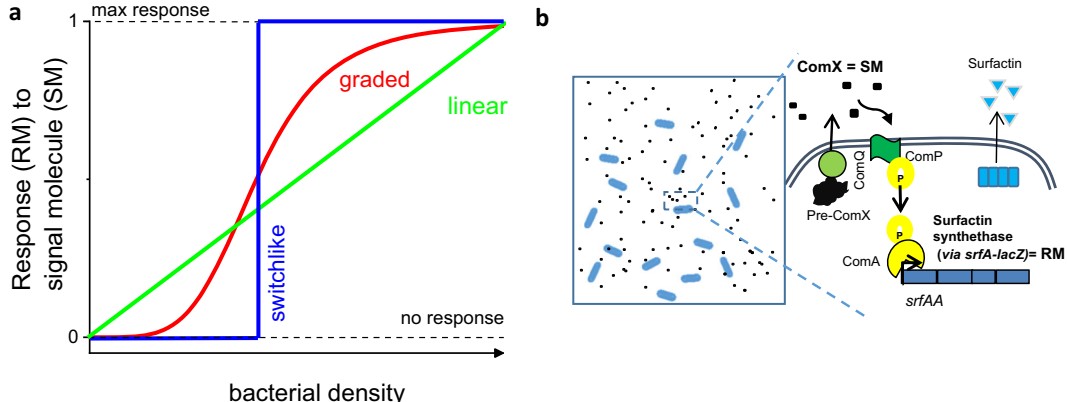

**Fig. 1 The modes of the signal molecule concentration-dependent response as a function of bacterial density. a** A communication system, where the response is linearly dependent on bacterial density tracks the bacterial density, however, there is no threshold density at which one could define the quorum. The switchlike transition from no response to max response describes an ideal quorum sensing system where upon reaching a critical concentration the QS response is attained. The less ideal, but anyway non-linearly controlled by the bacterial density, is the graded QS response. **b** The type of response mode in *B. subtilis* ComQXPA communication system where the peptide ComX activates the response, which is the transcription of genes encoding the response molecule (RM) surfactin synthethase, is unknown.

and experimental approach. In particular, we provide data on the time-dependent dynamics of (i) the concentration of the signal molecule SM (i.e., ComX), (ii) the cell density $N$, (iii) the critical SM concentration to elicit minimal quantifiable response (lower limit of response-LLR), (iv) the response (RM) to the SM, represented by the expression of the *srfA* operon encoding the surfactin synthetase, (v) the population distribution of signal molecule producers and signal responders.

Our results show that the ComQXPA communication system functions as a true QS system that adopts a switch-like dynamics, which is achieved by linking ultra-sensitive encoder module (signal production) with ultra-sensitive response module (signal response). The non-linear increase in signal molecule synthesis is coupled to the growth rate and oxygen concentration. We also show that signal production and response are predominantly spread over the entire population with limited heterogeneity. A very low concentration of costly signal molecule is sufficient for triggering the QS response, which can explain the observed low metabolic cost in signal molecule production.

## Results

### The concentration of signal molecule ComX increases by the square of bacterial density.

In order to determine the dynamics of signal molecule (ComX) production, we have used experimental and mathematical modeling approaches. We quantified the ComX concentration over time in the spent medium of PS-216 ($\Delta comP$) with the biosensor strain BD2876 (for strain-description see Supplementary Table 1), which produces β-galactosidase in response to the exogenous addition of ComX[34]. The assay included proper controls and calibrations to assure the biosensor-derived ComX concentrations are accurate (for details see Materials and methods). We found that the ComX concentration correlated positively with population density of PS-216 ($\Delta comP$) and remained constant at 10 nM after entering the stationary phase (Fig. 2a). Importantly, the representation of ComX concentration versus cell density (OD650) (Fig. 2b) showed a non-linear trend between the two parameters. The experimental data were fitted by an allometric function:

$$SM(t) = aN(t)^b \qquad (1)$$

Where $SM(t)$ is a signal molecule (ComX) concentration in time, $N(t)$ is bacterial cell density in time, expressed as optical density of the bacterial suspension (OD650). The fitting results for parameters $a$ and $b$ were (9.6 ± 0.6) nM a.u.$^{-2.09}$ and 2.09 ± 0.10, respectively. The value of parameter $a$ means that at OD650 = 1.0 a.u., which corresponds to the stationary growth phase in our experimental conditions and the bacterial density of $4 \times 10^8$ cells mL$^{-1}$, the ComX concentration is about 10 nM. In the early exponential growth phase the concentration was about 0.1 nM. Considering parameter $b$, the value obtained (2.09 ± 0.10) indicates that the ComX concentration increases by the square of bacterial density. This means that with increasing population density, the SM concentration (ComX) increases by the second power, while the amount of ComX per cell increases linearly. This relationship suggests that ComQXPA has an ultra-sensitive encoder module[9], where signal molecule production is very sensitive to cell density. The same mathematical relationship can be obtained by assuming that SM production rate per cell corresponds to the product of a specific cell growth rate and a cell density (i.e., population growth rate, for details, see Supplementary Methods, Derivation of ComQXPA communication system model). The dependence of the SM production rate per cell on (a) cell density and (b) the specific cell growth rate can be seen as an alternative way to obtain the ultra-sensitivity of encoders, which is usually achieved by SM dependent positive feedback in many

QS systems[9]. This makes ComX a true indicator of population density, which also encodes information about the cell growth rate.

### The production of signal molecule ComX in native concentration does not present a substantial metabolic burden for the producer.

Peptide signal molecules (SM) used by Gram-positive bacteria are metabolically costly to produce[29]. We estimate here that single molecule of ComX produced by *B. subtilis* used in this study (pherotype 168) requires a considerable investment of 484 ATP units per single signal molecule (for calculation details see Supplementary Methods, Calculation of ATP requirements for synthesis of 1 SM, ComX, 168 pherotype). This drastically exceeds the estimated cost of typical Gram-negative bacterial QS signals, with butyryl-homoserine lactone, C4-HSL from *Pseudomonas aeruginosa* costing only 8 ATP units[29]. However, the concentration of stationary growth phase signaling peptide ComX is 100–1000 times lower in *B. subtilis* (10 nM, Fig. 2a) compared to the typical concentrations of AHLs released by Gram-negative bacteria[35–38]. This suggests that high cost per SM is buffered by low concentrations of SM, thereby reducing the fitness costs of SM production in peptide-based communication systems. In order to test the metabolic cost of ComX production, we first compared the growth curves of receptor-deficient PS-216 ($\Delta comP$), and signal and receptor-deficient PS-216 ($\Delta comQXP$) strains (Fig. 2c). The use of the strains without a receptor made it possible to separate the costs of signaling from the additional costs of the communication response.

Apparently, the maxima of growth curves of $\Delta comP$ and $\Delta comQXP$, and their slopes (corresponding to the growth rate divided by log 2) were almost identical: PS-216 $\Delta comP = (0.503 ± 0.008)$ h$^{-1}$ and PS-216 $\Delta comQXP = (0.496 ± 0.007)$ h$^{-1}$, suggesting that ComX production does not represent a substantial metabolic burden in the observed system (Fig. 2c). The more direct fitness comparison between PS-216 ($\Delta comQXP$) and ($\Delta comP$), was carried out through a competition assay between PS-216 ($\Delta comQXP$) and PS-216 ($\Delta comP$) (Fig. 2d). In line with results in Fig. 2c, ratio of ComX producers and ComX non-producers did not changed considerably throughout the experiment, suggesting negligible costs for signal production (Fig. 2d).

Next, we tested whether the absence of prudent SM production induces measurable fitness cost. To test this, we overexpressed *comX* from the P$_{hycomX}$ IPTG-inducible promoter (Supplementary Fig. 2a, b), which ensured the production of additional copies of ComX. As expected, the overproduction of ComX has a negative impact on the growth of *B. subtilis* (Supplementary Fig. 2a). The overexpression of ComX in *E. coli* had a similar negative fitness effect (Supplementary Fig. 2c). As it can be calculated from Supplementary Fig. 6b, the concentration of ComX in *E. coli* spent media was about 900 nM, corresponding to 200 nM a.u.$^{-1}$, which is about 20 times more than we have measured in *B. subtilis* (Fig. 1a). The above results indicate that the costs of ComX synthesis under the native production regime are very low and can only be evaluated under non-native overexpressing conditions.

### The ComQXPA communication system operates in strong correlation with the oxygen concentration.

As already mentioned, the SM production rate per cell in ComQXPA is not controlled by the SM dependent positive feedback loop, but by cell density and specific cell growth rate (population growth rate), (eq S5-S6). The question is how bacteria then sense cell density and specific growth rate, which accelerate signal production. One of the key factors for the growth rate of *B. subtilis* is

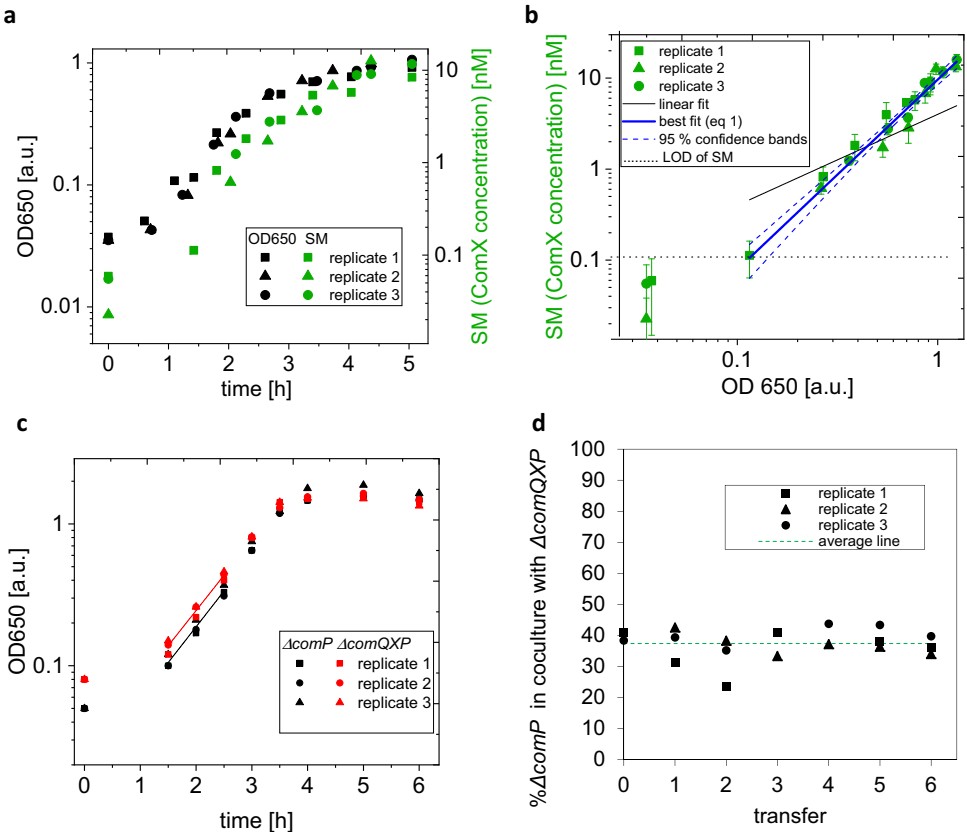

**Fig. 2 The accumulation of signal molecule (SM) during the growth of *B. subtilis* and fitness cost of SM production. a** The growth curve (OD650) of *B. subtilis* PS-216 Δ*comP* (no signal receptor) producing SM (ComX) that is accumulating in the growth medium of fermenter working in the batch mode of *n* = 3 biologically independent replicates is presented; **b** The experimental data *n* = 3 biologically independent replicates where the data ≥ limit of detection of SM was fitted by Eq. (1); the error bars for SM concertation are standard errors calculated from 8 technical replicates for each biological replicate; **c** The comparison of growth curves of *B. subtilis* PS-216 with no signal molecule receptor (Δ*comP*) and no signal molecule production and receptor (Δ*comQXP*) of *n* = 3 biologically independent replicates; the OD650 at *t* = 0 h was corrected with respect to the measured CFU of the inoculum. The slopes of the fitted lines in **c** correspond to the growth rate divided by log 2; the exponential phase points in the most reliable OD650 region (>0.1 a.u. and <0.7 a.u.) were considered. The slopes do not differ significantly (*P* = 0.32): Δ*comP* = (0.496 ± 0.007) h⁻¹ and Δ*comQXP* = (0.503 ± 0.008) h⁻¹. **d** The same strains grown in co-culture; each time OD650 reached 0.6 a.u. the co-culture was transferred to the fresh medium; *n* = 3 biologically independent experiments were performed and each time 6 of 7 transfers were checked for CFUs of both strains.

environmental oxygen content[39], which is believed to determine the survival strategies of this species[40]. It was recently shown that surfactin, which is directly related to the response in ComQXPA (RM in this work) becomes critical for *B. subtilis* when oxygen is low[41].

First, we did not allow any aeration of the batch culture, i.e., the oxygen supply to the growing culture was limited by diffusion of air through air filters on the inlets of the incubator. We monitored changes in oxygen concentration during growth in batch culture and observed an almost perfect negative correlation between the growth curve and the dissolved oxygen concentration (Fig. 3a, Supplementary Fig. 3). The strongest decrease in dissolved oxygen in the medium occurred during the exponential growth phase, exactly when the population growth rate reached its maximum. When spent medium of PS-216 wt was tested by the ComX biosensor BD2876 (Δ*comQ*, *srfA-lacZ*), we could measure the significant response (for *t* > 1.75 h, *P* < 0.008) by the biosensor that increased with the growth of the culture (Fig. 3c), indicating ComX is being produced. This agrees with Fig. 2a, where we quantified the produced ComX in the spent medium of PS-216 (Δ*comP*, producing ComX, but not responding to ComX). As expected for the proper ComX biosensor, it barely responded to the tested spent medium with no ComX (PS-216 Δ*comQ* spent medium) and strongly responded to the same medium when

purified ComX was added (Supplementary Fig. 4) confirming the ComX is the major factor being measured by the biosensor BD2876.

Next, we assessed the signal production in batch culture, where we assured continuous oxygen saturation. Under this condition the negative relationship between population size and dissolved oxygen concentration is broken. Surprisingly, oxygen saturation eliminated ComX production, which can be seen by very low biosensor BD2876 response that is indistinguishable from the spent medium with no ComX (PS-216 Δ*comQ* spent medium) even in the late-stationary growth phase (Fig. 3d). This result indicates that the ComQXPA communication system has lost its functionality, when there is no 'natural' oxygen gradient, and that the oxygen content can be used as an indicator of cell density and growth rate.

**The response model shows that the response of the cells to ComX is non-linear.** In our model, the expression level of the *srfA* operon serves as a measure for the response (RM) to the signal molecule SM, represented by ComX. To study how RM depends on SM we evaluated promoter activity of *srfA* in the *B. subtilis* PS-216 (Δ*comQ*, P*srfAA*-*yfp*), which carries the markerless deletion of *comQ*[42] and is therefore signal-deficient. Response level was assessed by incubating the PS-216 (Δ*comQ*, P*srfAA*-*yfp*)

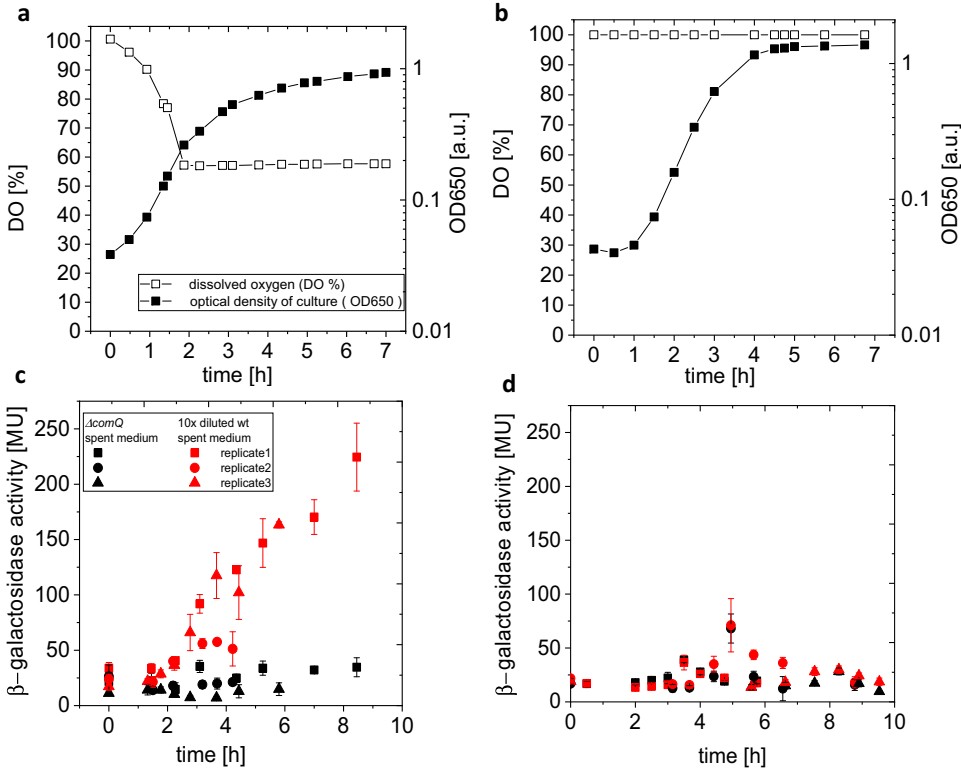

**Fig. 3 Influence of O₂ on the presence of SM (ComX). a**, **c** The strain PS-216 wt was grown in the fermenter batch system where oxygen supply was limited or **b**, **d** supplied to the saturation. **a**, **b** The growth was monitored by OD650 and oxygen saturation was followed by a polarizable electrode. **c**, **d** The samples of spent medium were periodically taken to test the presence of ComX via β-galactosidase activity of the ComX biosensor BD2876. The biosensor was incubated in the fresh CM medium supplemented with either spent medium of the ΔcomQ strain (no ComX, negative control, black color), or supplemented with the wt strain spent medium 10 times diluted by spent medium of the ΔcomQ strain (red color); the spent medium of the ΔcomQ strain was obtained in the parallel batch system; n = 3 biologically independent experimental replicates are presented with error bars representing SD of 8 technical replicates. The comparison to positive control is given in Supplementary Fig. 4.

for 4 h in the presence of different ComX concentrations, which was the only factor that varied in this experiment. The response level was expressed as Yfp fluorescence per cell, normalized to the maximum response, $W_{max}$, which gives a relative measure, $W$ (SM), of how strongly the cells respond to ComX and this is shown as a function of the exogenously added ComX in Fig. 4a. The response to SM was sigmoidal. In order to check whether the response curve had reached the final shape after 4 h of biosensor incubation, we performed the same experiment, but incubated biosensor with ComX for 3 or 6 h, respectively. As can be seen from the comparison of Fig. 4a with Supplementary Fig. 5, the response curve has not changed after extending the incubation over 4 h. We have therefore taken the 4 hours response curve (Fig. 4a) as a reference for further communication system analysis. The sigmoidal functions can typically describe the relationship between transcription factors and promoter activities, and can be modeled by the Hill equation[43,44]. In the case of ComQXPA the ComX dependent transcription factor ComA-P acts directly on the $P_{srfAA}$ promoter and induces its activity as the response to the signal. Assuming a linear relationship between ComX concentration (SM) and the active ComA-P one can expect that the experimental data can be fitted by the Hill equation:

$$W(SM) = \frac{W_{max}SM^n}{Km^n + SM^n} \quad (2)$$

SM is ComX concentration and Km is the ComX concentration at which half of the maximum response is achieved; n (Hill

coefficient) describes the cooperativity among transcriptional activators. Successful fits indicated by the low reduced $\chi^2$ (see Supplementary Table 2), show that the biosensor sensitivity is maximized at 3–5 nM of ComX (Km), while the highest response value is reached at around 10 nM of the ComX. n > 1 values obtained for all fits indicate positive cooperativity (i.e., ultra-sensitivity[14]) in the binding of the transcriptional factor ComA to the srfA promoter[43,44]. This agrees with the research showing that two molecules of the ComA homodimer cooperatively bind to the two promoter regions located upstream of the RNAP binding sites of srfA[13,20,45]. The inactivation of the second promoter region decreases the promoter activity of srfA by 100-fold (ref. [13]), which underscores the importance of the second binding region, explains n ≥ 2 and the sharp increase in srfA promoter activity with ComX concentration. In addition, we show here that the critical concentration of ComX required to induce quantifiable response (designated here as lower limit response (LLR)) is 0.2–0.5 nM. These results, therefore, suggest that the response per cell depends cooperatively on the ComX concentration and that the cells respond to very low concentrations of ComX.

**Fully functional ComQXPA communication system does not require a positive feedback loop–the validation of the ComQXPA communication system.** The response curve in Fig. 4a is a function of the SM concentration only. In the more natural setting (i.e., during growth) the cells encounter growth-dependent changes in SM concentrations as well as changes in bacterial density and growth rate over time. We have therefore

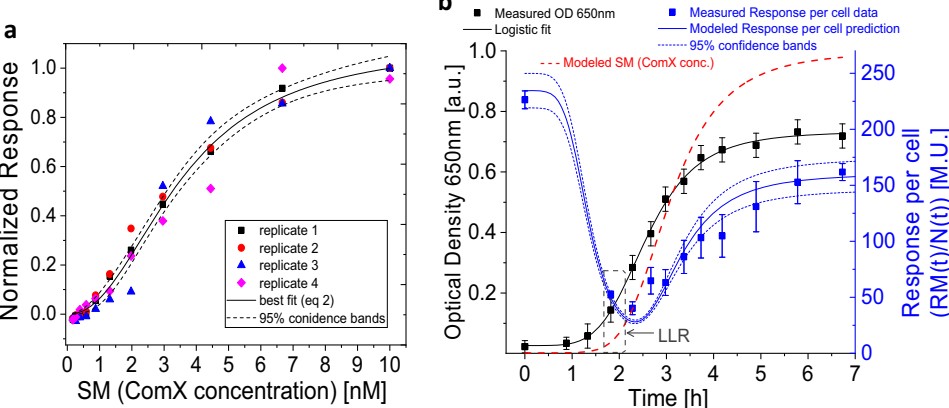

**Fig. 4 Influence of oxygen on the presence of SM (ComX). a** Signal molecule deficient *B. subtilis* PS-216 (*ΔcomQ*, *P_srfAA-yfp*) was incubated in the presence of SM for 4 h and the maximum normalized response was determined from the activity of the *srfA* promoter. $n = 4$ biologically independent replicates were performed. Best, concatenated fit to the model in Eq. (2) is presented together with 95% confidence level. **b** the logistic fit to one of the three growth curves ($n = 3$, biologically independent replicates) measured as OD650 of the culture *B. subtilis* PS 216 (*srfA-lacZ*) producing signal molecule, SM that accumulated in the growth medium of batch system is shown. The response per cell data, obtained as the β-galactosidase activity of *srfA* promoter of *B. subtilis* PS-216 (*srfA-lacZ*) was fitted by Eq. (3), ($R^2 > 0.99$). The time interval at which SM concentration is high enough to cause the measurable response, i.e., lower limit of response (LLR) as predicted from data in experiments in (**a**) is given as dashed window in (**b**). One of the five qualitatively and quantitatively similar experimental results is presented. Error bars represent SD of 8 technical replicates. For fits of additional replicates and data variability refer to Supplementary Tables 3 and 4.

asked whether the response curve based on the modeling and results presented in Figs. 2b and 4a could fit the response data in the SM producing and responding strain exposed to changes in these three parameters.

We cultivated the SM producing and responding PS-216 strain carrying the response reporter (*PsrfA-lacZ*) in a large volume bioreactor system (Fig. 4b). This allowed sterile sampling of spent medium and cells (for response quantification) at several time points, without affecting growth conditions. Immediately after the inoculation of the fresh medium by overnight culture the β-galactosidase activity of PS-216 (*PsrfA-lacZ*) was high. We assumed that this was a consequence of the accumulation of the expressed *PsrfA* reporter (β-galactosidase, RM) during the overnight growth. As a consequence of the dilution of the intracellular β-galactosidase (RM) due to cell division, the activity of the β-galactosidase decreased sharply after 2 h incubation (Fig. 4b). Simultaneously, as predicted by (Eq. 1), the concentration of SM (ComX) in the medium was increased exponentially during growth, and soon reached a critical concentration to activate the *srfA* promoter. In particular, as elucidated by the fits of (Eq. 2) to the data in Fig. 4a, the lower limit of the response (LLR) is reached shortly before upturn of the cell response curve in Fig. 4b. At this point the culture is in exponential growth phase at the cell density of 3 to $8 \times 10^7$ cells mL$^{-1}$. The steep slope of the response curve indicates that the rate by which the response molecule (RM) is synthesized now exceeds the dilution due to the cell division rate. From now on, the response per cell correlates approximately linearly with OD650, suggesting a strong coupling to cell growth. Taking these facts into account and considering that the response molecule (RM) concentration is sensitive to the concentration of the signal, SM, (Eq. 2) and that SM can be expressed in terms of cell density (Eq. 1), the concentration of a response molecule per cell, *RM(t)/N(t)*, can be analytically described (see also Supplementary Methods, Derivation of ComQXPA communication system model) as:

$$\frac{RM(t)}{N(t)} = \frac{RM0}{N(t)} + \frac{RM1(t)}{N(t)} \qquad (3)$$

where *RM0/N(t)* is the response per cell of overnight culture, i.e.,

the overnight accumulated β-galactosidase. The second term, *RM1(t)/N(t)* accounts for the synthesis of the β-galactosidase after inoculation of a fresh medium and comprises the parameters describing the sensitivity of the response to a signal molecule, *Wmax*, *Km*, *n* (Eq. 2), the signal production, *a* (Eq. 1), cell density, *N(t)* and proportionality constant, *k* that gives the magnitude of the response per cell when the potential to respond to the signal is maximally fulfilled (i.e., at *Wmax*) and the specific growth rate is 1 h$^{-1}$. The definition of *RM1(t)* is given in Supplementary Methods (eq S11). Note that for the derivation of Eq. (3) we assumed no degradation of SM occurs, as our experiments suggest SM was stable under the conditions studied (Supplementary Figs. 6a and 7, see also Supplementary Methods, Derivation of ComQXPA communication). All the parameters in (Eq. 3), except *k* in *RM1(t)* and *RM0*, were taken as constants obtained in the independent experiments by fits of (Eq. 1) and (Eq. 2). With *k* and *RM0*, as the only fitting parameters, we applied the mathematical model in (Eq. 3) (for details of the model equation see Supplementary Methods, Derivation of ComQXPA communication system model) to fit the experimental cell response data (Fig. 4b). The successful fit (see Supplementary Table 4 for details) indicates that the relationship assumed among cell density, cell growth, signal concentration and response in (Eq. 3) is valid and yields (760 ± 120) M.U. for *k* and (5.5 ± 1.5) M.U. a.u. for *RM0*. Again, we did not need to incorporate the SM feedback loop into our model, which is consistent with published results suggesting that this communication system lacks a feedback loop[10–13].

**The ComQXPA dependent signaling and response at the cellular level.** So far, we have focused on the population averages, which is a traditional approach in studies on microbial communication systems[17,46]. We here report results on communication dynamics of *B. subtilis* at the single cell level using fluorescence-based molecular tools. This approach provides the means to track temporal changes in expression of genes involved in signal synthesis (signaling) and in response and thus provides the insight into a phenotypic heterogeneity within the population.

We used the double-labeled fluorescent strain *B. subtilis* PS-216 (*comQ-yfp*, *srfA-cfp*), in which fluorescent reporters were

fused to the *comQ* and *srfA* promoters. The two genes code for the ComX signal-processing protein and the communication-activated operon, respectively. Since *comQ* and *comX* share the same promoter and their genetic sequences often overlap[15] expression level of *comQ* corresponds to the expression level of *comX*. The fluorescence of individual cells was observed under the microscope in different growth phases and quantitatively analyzed (Fig. 5).

The observed expression pattern for the signaling gene (*comQ-yfp*) follows lognormal distribution (Fig. 5g). A small number of outliers in the *comQ* expression (on average, 10x brighter than the majority) were easily detected in the qualitative image analysis (Fig. 5a). These hyperproducers were not present in the overnight culture and began to occur during exponential growth, after 1-hour incubation in fresh medium, (Fig. 5d). In general, hyperproducers accounted for about 0.1–1% of the population, and their frequency increased during the first 6 hours. These data suggest that bulk of the ComX is nevertheless produced by the majority of the population as expected for the true QS communication systems. The contribution of the *srfA* hyperproducers to the total surfactin production is even less pronounced since their occurrence did not exceed 0.1% of the population (Fig. 5b, d).

The most heterogeneous expression of the communication signaling gene (*comQ-yfp*) was observed in overnight culture, immediately after its transfer to the fresh medium (Fig. 5g, Supplementary Fig. 8a), but hyperproducers where not detectable at this time (Fig. 5d). Once the cells begun to divide, the distribution shifted to lower fluorescence intensities with a simultaneous decrease in heterogeneity, but from 3 to 4 h onwards single cell fluorescence gradually increased, along with an increase in population heterogeneity (Fig. 5e, g, Supplementary Fig. 8a). This suggests that the expression rate is now higher than the division rate (i.e., the production overpowers the dilution due to cell division). A similar pattern was observed in the communication response (*srfA-cfp*) (Fig. 5f, h, Supplementary Fig. 8b) with two major differences. The minimum level in *comQ-Yfp* fluorescence was reached 1 h later than *srfA-Cfp* fluorescence, which suggests the expression of ComX is in first hours low compared to the *srfA* expression. Nevertheless, the entire cell population, with the exception of hyperproducers, which represented only a fraction of the *comQ/srfA* expressing cells, followed unimodal lognormal distribution expression pattern. This suggests that the ComQXPA communication phenomenon in *B. subtilis*, at least under the conditions in our experiment, is not restricted to individuals and can be studied at the population level, i.e., the averages represent well the population.

The *comQ-yfp* and *srfA-cfp* expression co-localization analysis (Supplementary Table 5) reveals the correlation coefficient of about 0.5, which is significantly ($P = 0.01$) higher than in the overnight culture. The presence of the correlation suggests that on average, cells that produce the signal more intensively, also respond to signal more intensively, supporting the idea of self-sensing[47]. However, the correlation coefficient strength was only moderate, suggesting that sensing of the external signal (sensing-of others) still works as expected for a typical QS system.

**The induced response in ComQXPA communication system is graded and almost switch-like.** The perfect QS system does not produce a response until the threshold bacterial density is reached and then immediately switches to a full response. This minimum to maximum transition may be either a perfect switch or a graded induction. By combining the information from Figs. 2b and 4a in the form of eq S9 results in the normalized ComQXPA response curve as a function of bacterial density (Fig. 6a) that resembles a

graded switch like induction (compare to Fig. 1a). The perfect switch like communication system is unrealistic, because it requires that all the cells are perfectly synchronized and immediately switch to maximal response, leaving no time for the adaptation to the signal stimuli. It is reasonable to expect that for the true quorum sensing system (QS) most of the response has to occur within the same generation of dividing cells ($n_{gen} < 1$). As can be seen in Fig. 6b, this is true for the ComQXPA communication system (our case), which achieves 50% of the response within the same generation of dividing cells ($n_{gen} \approx 0.7$). On the other hand, if the communication system lacks the ultra-sensitivity in either the signal production (encoder module) or response production (decoder module), the achievement of 50% response shifts well over the same generation of dividing cells ($n_{gen} \approx 1.4$ to $1.6$), extending the cell density and time needed for substantial response to occur.

## Discussion

The variable peptide and isoprene moiety of ComX signaling molecule[19] enables this molecule to be intra-species specific[18,33]. High specificity has its price—our estimation of ComX molecule cost is 484 ATP, considerably more than 8 ATP for acyl-homoserine lactones (AHL)[29] based communication systems. It was shown that cost of AHL synthesis is high enough to experimentally measure its fitness cost[31], which was for the peptide SM undetectable in our assays. The most obvious reason for this is that communication system in *B. subtilis* operates in nM concentration range (10 nM a.u.$^{-1}$, i.e., 10 nM of SM concentration per unit of culture optical density), whereas SM of other communication systems are often in µM concentrations[23]. This includes the study showing the measurable fitness cost of AHL, where AHL concentration was 20 µM a.u.$^{-1}$ (ref. [31]). Therefore, our results imply that the problem of high cost per molecule of ComX peptide was mitigated by *B. subtilis* by deploying communication system that operates at very low concentrations.

The prerequisite for a well-functioning quorum sensing system is the ability of the signal molecule concentration to follow cell density, which is the role of encoder module. Drees et al.[9] theoretically predict the ComX encoder module consisting of the *comQ* and *comX* gene, to be an Ideal class. In this class signal molecule concentration increases linearly with cell density, with the exponent b in Eq. (1) being 1, implying a constitutive synthesis of SM (see Supplementary Methods, Derivation of ComQXPA communication system model). However, our results indicate that SM (ComX) concentration in *B. subtilis* increases with the square of cell density, which classifies the encoder module as the Ideal ultra-sensitive ($b > 1$), meaning that the increase in signal molecule concentration is very sensitive to the increase in cell density and that SM production must be controlled by an additional factor. This is generally represented by a positive signal autoregulation[48], which is missing in ComX encoder module[9–13,49]. Our modelling suggests that the rate of ComX production depends on specific cell growth rate and cell density (see eq S1-S6), which means that *B. subtilis* controls the production of SM (ComX) by sensing others (cell density) through an alternative mechanisms. According to our data this mechanism involves the detection of dissolved oxygen. First, *B. subtilis* produces ComX only in the oxygen diffusion limited medium, but not in an oxygen saturated medium, independent of cell density and growth rate. Second, the concentration of dissolved oxygen (DO) decreases sharply when the population growth rate increases and oxygen consumption exceeds supply (Supplementary Fig. 3). This occurs between 1.5 and 2.5 h after inoculation, which coincides with the increase in signal production and the LLR window, the earliest measurable

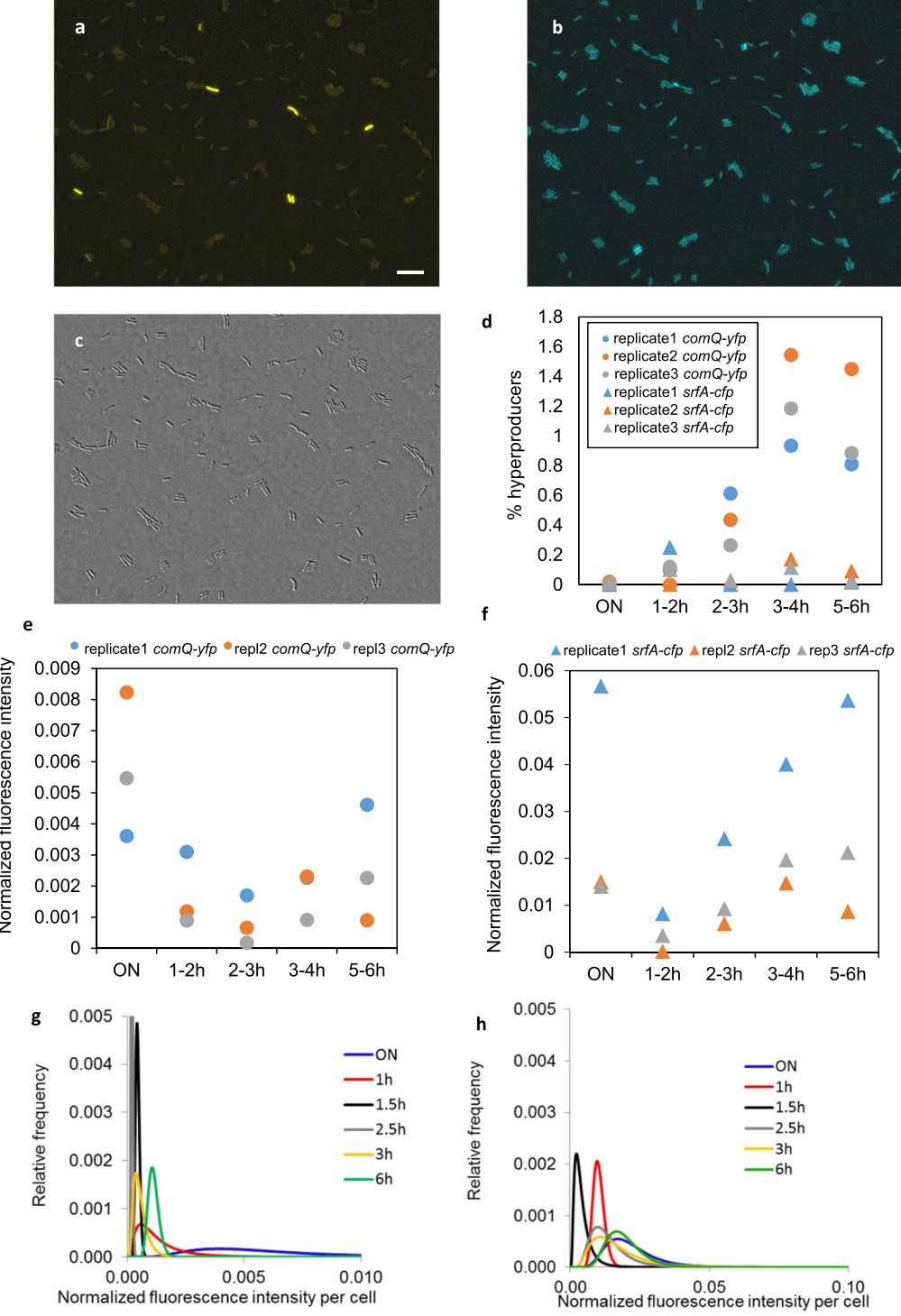

**Fig. 5 Single cell quantitative fluorescence microscopy of *B. subtilis* PS-216 (*comQ-yfp, srfA-cfp*).** The fluorescence microscopy images were taken periodically during incubation of *B. subtilis* PS-216 (*comQ-yfp, srfA-cfp*) in a batch fermenter by the YFP filter (**a**), CFP filter (**b**) or DIC (**c**). The example YFP and CFP images, taken after 3 h of incubation were pseudo-colored. The scale bar represents 10 μm. $n = 3$ biologically independent experiments were performed. **d** % of population of cells that are hyper-expressing *comQ-yfp* or *srfA-cfp* is depicted. Gene expression level determined by single cell fluorescence microscopy was measured as Na-fluoresceinate standard normalized mean fluorescence intensity per cells expressing *comQ-yfp* (**e**) and *srfA-cfp* (**f**); ON is the overnight culture. One of the three qualitatively similar cell distributions is shown in (**g**) and (**h**) for *comQ-yfp* and *srfA-cfp*, respectively; areas under the curves are the same in all time points. For additional replicate see Supplementary Fig. 8.

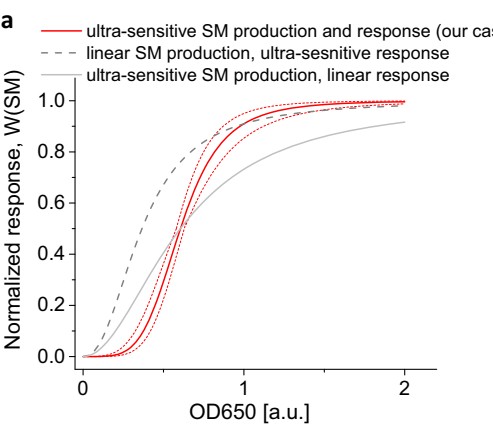
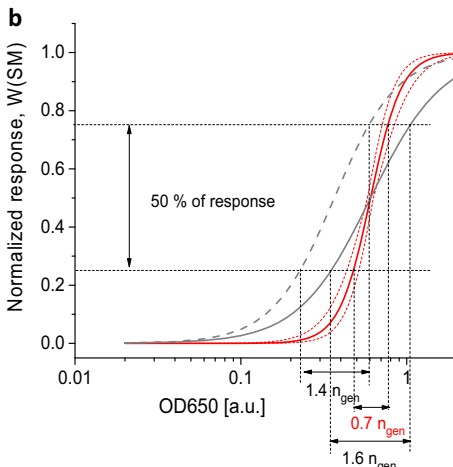

**Fig. 6 The importance of the quadratic dependence of signal molecule (SM) on bacterial density (ultra-sensitive SM production) and cooperativity in response to SM (ultra-sensitive response).** The maximum normalized response curve (red line), eq S9, as obtained on the basis of experimental data in Fig. 2b, Fig. 4a and presented as a function of bacterial density expressed as linear (**a**) or logarithmic (**b**) optical density, OD650. The conversion of SM concentration to OD650 was performed by Eq. 1, where SM increased, as measured, by the square of bacterial density. The shown error (95% CI) of eq S9 (red dotted line) is due to the uncertainties in values of parameters of Eqs. 1 and 2; A theoretical case, where SM production linearly depends on bacterial density (gray dotted line) or shows a linear response (gray line). Note the much wider window of response for linear SM production or response compared to ultra-sensitive production and response, where 50% of response occurs within the same generation of growing cells ($n_{gen} = 0.7$). Therefore, only the ultra-sensitive-response (decoder module) and ultra-sensitive signal molecule production (encoder module) operating in the same communication system give the switch-like response, necessary for the true quorum sensing response (red line).

response to the signal (Fig. 4b) followed by the upturn of the response curve. Our results thus underline oxygen as an independent factor influencing signal production and are in good agreement with experiments by Ghirbi et al.[50], who observed a significant decrease in *B. subtilis* surfactin production, when DO was increased from 40 to 60%. The recent research reveals that surfactin promotes growth in early stationary phase by enhancing oxygen diffusion and that surfactin maintains viability upon oxygen depletion, which becomes critical at high cell densities[41]. This explains the factors influencing the production of ComX (SM in our study), a major signal for surfactin synthethase expression (RM in our study). Cells need surfactin, when their numbers expand quickly and oxygen is dropping. The mechanism on how oxygen modulates the ComX production remains to be elucidated, however, Ohsawa et al.[16] showed that the expression of the *comQXP* locus decreases with high levels of superoxide ($O_2^-$). Since ROS production-rate is proportional to collision frequency of oxygen and redox enzymes, the rate of $O_2^-$ and $H_2O_2$ formation inside the cells depends directly on the oxygen concentration in the extracellular environment[51–54]. Although it might be tempting to conclude that in this case sensing superoxide is equivalent to sensing oxygen, it is important to note that the superoxide concentration is the "private" property of the cell, while oxygen concentration is shared by all cells and can carry the population-related information.

In general, a bacterium could achieve the same level of graded, nearly switch-like induction as in our case, by deploying the SM production with even higher level of ultra-sensitivity, with $b > 2$ in Eq. 1, (and with linear response to SM), like in the case of a positive feedback regulated systems[9], where SM stimulates its own production. Such systems, typical for AHLs production, can rapidly drive SM production to very high concentrations and thereby increasing substantially the costs. This could be detrimental for a bacterium, especially if the cost per SM is high, like in the case of peptide ComX. To prevent this scenario *Bacillus* omits SM feedback loop thereby lowering the ultra-sensitivity (to $b = 2$ in Eq. 1) in SM production. The partial loss of ultra-sensitivity in SM production is, however, compensated by the

ultra-sensitive response ($n = 2.3$, Eq. 2). Therefore, by distributing the ultra-sensitivity between the encoder and decoder, the ComQXPA communication system yields a response that has a steep enough transition from minimum to maximum response level to function as a true quorum sensing communication system. This ultra-sensitive and economic quorum sensing system, used by the entire cell population, relies on the oxygen concentration to adjust the signal production to the growth rate and the cell density.

## Methods

**Bacterial strains.** Strains used in this study are listed in Supplementary Table 1. To obtain strain BM1042 the PS-216 *srfA-cfp (cat)*[55] was transformed with plasmid pKB72 isolated from *E. coli* strain KTB360[11]. To obtain the strain BM1297, PS-216 *srfA-cfp (cat)*[55] was transformed with the DNA isolated from strain DL954 (*amyE:: PcomQ-yfp (spec)*)[56]. The strains are available from the authors upon request.

**Bacterial growth in batch fermenter system.** To measure concentration of the signal molecule (SM), i.e., ComX, and response to the signal molecule (RM), i.e., *srfA* promoter activity, during growth of *B. subtilis* the Minifors (Infors, AG, Switzerland) fermenter system operating in batch mode was deployed. Overnight cultures were grown in competence medium[57] (CM) supplemented with L-histidine, L-leucine, and L-methionine (50 μg mL$^{-1}$). Prior to fermenter inoculation, the cells were washed twice by replacing the spent medium of overnight culture with SS buffer[58]. Cell suspension was inoculated into 1.2 L of fresh CM (2% inoculum). Incubation was performed at 37 °C with mixing at 700 rpm and the closed off air flow. In instances where we tested $O_2$ impact on SM concentration the medium was saturated with $O_2$ by ensuring a supply of compressed air at a flow rate of 1 L min$^{-1}$. The dissolved $O_2$ was measured using the $O_2$ polarized electrode InPro 6820 (Mettler Toledo). No pH correction was performed or anti foaming agents were added.

**Biosensor based quantification of signal molecule (ComX) concentration.** We have developed a biosensor-based method to measure the concentration of SM (ComX) in the spent medium of *B. subtilis* at different growth times. The biosensor BD2876 (Supplementary Table 1) does not produce SM (ComX), because it lacks the essential enzyme ComQ that is required for ComX maturation. However, it can respond to exogenous ComX via the receptor/response regulator pair ComP/ComA that trigger the *srfA* promoter during the incubation of the biosensor in the medium containing ComX. The activity of this promoter can be measured via β-galactosidase reporter and correlated to ComX concentration. Biosensor test provides two data sets: (i) β-galactosidase activity as a function of relative SM concentration in spent media (unknown concentration of ComX) and (ii) β-

galactosidase activity as a function of known SM concentration. After fitting these data to the Hill equation (Eq. 2) we obtained a sample and the calibration curve, respectively. The concentration of SM in spent medium in absolute units was calculated based on the comparison of the two curves (for details see Supplementary Methods, ComX concentration measurements) by considering the linear part of the sigmoid curve. In this way a possible saturation and consequent underestimation of ComX concentration was avoided. An example of the curves is given in Supplementary Fig. 1. The obtained ComX concentration for a given time point in spent medium has experimental error that includes random errors of calibration and sample curve. Importantly, we took the following measures to assure that in our bioassay the calibration curve and sample curve differed only in the ComX concentration, which is necessary to accurately quantify the unknown ComX concentration in the samples: (i) the samples with unknown concentration of ComX were always spent media of *B. subtilis* PS-216 (*ΔcomP*), therefore, this spent media did not contain any potential products that result in ComX sensing (ii) for sample curve *ΔcomP* spent medium was diluted by *ΔcomQ* spent medium, obtained by *B. subtilis* PS-216 (*ΔcomQ*), that in addition to the absence of products resulting from the ComX presence also does not contain ComX itself (ComQ is essential enzyme for ComX production). Therefore, composition of concentration series in standard curve differed only in ComX concentration, the other factors that could potentially influence biosensor response, as possible presence of extracellular peptide signals from the Rap-Phr family were kept constant. (iii) The same *ΔcomQ* spent medium was used in calibration curve, where known amounts of purified ComX were added to *ΔcomQ* spent medium, which means that the only factor changing in concentration series for calibration curve was ComX concentration. (iv) The *ΔcomQ* and *ΔcomP* spent media were obtained always in parallel experiments under same exact growth conditions and at the same OD650 of *B. subtilis* PS-216 *ΔcomQ* and *ΔcomP* strains. (v) For each time point in the growth curve of *B. subtilis* PS-216 *ΔcomP* a microtiter plate containing samples for calibration and sample curve was assayed by biosensor BD2876 under the same growth conditions. (vii) β-galactosidase test performed on biosensor BD2876 was done simultaneously, under the same conditions, for calibration and sample curve. In addition, the accuracy of the test was enhanced by adding BSA to block non-specific interactions of partly hydrophobic ComX molecule to purified (standard) ComX, *ΔcomQ* and *ΔcomP* spent medium. The purity of the HPLC quantified ComX was additionally tested for purity and identity on MS, where the presence and purity >90% of ComX was confirmed (see Supplementary Methods, Quantification and isolation of purified ComX).

**Fitness cost of signal production.** To determine the fitness cost of signal production we compared the growth dynamics of signal negative *B. subtilis* PS-216 (*ΔcomQXP*) and receptor negative PS-216 (*ΔcomP*) strains. Bacteria, grown as overnight cultures in LB medium, were inoculated into fresh CM medium (2% inoculum) and incubated at 37 °C, at 200 rpm for indicated times. The growth was followed by measuring optical density at 650 nm. To further assess the fitness cost of ComX synthesis, *ΔcomP* (does not respond to the signal) and *ΔcomQXP* (does not produce ComX nor responds to it) strains were grown overnight in LB medium and then cultured as 1:1 co-culture (based on OD650 of overnight culture at start-transfer 0) in fresh CM medium at 37 °C, 200 rpm up to mid. exponential phase. To amplify potential fitness differences between the two strains the exponential growth phase was prolonged by replacing half of the culture volume with fresh CM medium every 60 min. Relative CFU of *ΔcomP* and *ΔcomQXP* strains were accessed at the start (transfer 0), and every 60 min from the start of mid. exponential phase (transfer 1-6).

To examine metabolic burden of signal hyper-production, we compared the growth curves of strain PS-216 wt and its isogenic recombinant, PS-216 (*PhyComX*), carrying additional copy of ComX under the IPTG inducible promoter. The overnight LB cultures of the PS-216 wt and PS-216 (*PhyComX*) were transferred into CM medium containing 0.2 mM IPTG and incubated at 37 °C, 200 rpm. The growth was monitored by optical density at 650 nm (OD650). The presence of different ComX concentrations was determined by measuring the activity of spent medium using the biosensor strain (BD2876) and the β-galactosidase assay (see Supplementary Methods, β-galactosidase assay).

To examine metabolic burden of ComX overproduction in a synthetic setup we compared the growth curves of *E. coli* ED367, carrying a plasmid (pET22b) with the IPTG inducible *comQcomX* genes[34], under non-inducing and inducing conditions. The overnight LB culture of *E. coli* ED367 was transferred into M9 medium containing ampicillin (100 μg mL⁻¹) and grown at 37 °C, 200 rpm. The growth was monitored by OD650. To induce overexpression, the cultures were supplemented with IPTG (0.2 mM) when the OD650 reached 0.6 a.u.

**Quantification of the signal induced population response.** To quantify a response (RM) to different signal molecule (SM) concentrations at the population level we used the signal deficient *B. subtilis* PS-216 (*ΔcomQ P_srfAA-yfp*), which carries the *comQ* marker-less deletion and the *srfA* promoter tagged with the Yfp fluorescent reporter (Supplementary Table 1). The response to different concentrations of the added SM (ComX) was quantified by fluorescence microscopy.

To get enough ComX the original growth medium for heterologous expression of ComX in *E. coli* ED367[34,55] was further optimized (see Supplementary Methods, Optimization of growth medium for heterologous expression of ComX in *E. coli*

ED367). A series of ComX dilutions (dilution factor 1.5x) was prepared by mixing 200 nM stock solution of ComX in SS buffer. To prevent non-specific binding of the ComX filter sterilized BSA was added[17] (final concentration of 50 μg mL⁻¹). A total of 5 μL of each dilution of ComX was then transferred to microtiter plate wells (96 Well Polystyrene Cell Culture Microplates, black, clear bottom, Greiner Bio One) with 95 μL fresh CM medium that contained inoculum (2%) of PS-216 (*ΔcomQ, PsrfAA-yfp*), 8 replicates per dilution. The plate was placed on a microtiter plate shaker (1 mm orbit, 1100 rpm) that was positioned in the humidified chamber to eliminate sample evaporation. To improve the homogeneity of the conditions (airflow, temperature) on the microtiter plate the microtiter plate lids were elevated by 2 mm relative to the surface of microtiter plate. The plates were incubated at 37 °C for 3, 4, or 6 h. The cells were then re-suspended and fixed in 200 μL of 1% agarose containing DAPI dye (6 μg mL⁻¹).

For quantitative analysis we used laser confocal fluorescence microscopy (Zeiss, LSM 800). The images were acquired by two laser channels—488 nm laser for acquiring fluorescence from Yfp protein and 405 nm laser to acquire fluorescence from DAPI dye. The acquired images were 1024 × 1024 pixels in size with 16-bit color depth at 20× optical magnification (objective 20×/NA 0.4, Zeiss) and were analyzed with Fiji-ImageJ[59] custom script[60]. Four images per well of microtiter plate were taken, each image contained about 350 cells. The measure for amount of Yfp on image was integrated density i.e., the product of mean fluorescence intensity and area taken by the cells. The response per cell was quantified as the ratio of image integrated density by the number of cells obtained from DAPI image.

To quantify the response (RM) at the population level to the SM during growth, where SM concentration is continuously growing, we used the signal producing and sensing strain *B. subtilis* PS-216 (*srfA-lacZ*) cultured in CM media in batch fermenter system. At different time points during growth 12 mL of culture were collected from the fermenter, OD650 determined, and centrifuged (5 min, 8000 g) to remove supernatant. Cell pellets were then stored at –20 °C for later analyses of β-galactosidase activity. For this purpose, cells were re-suspended in SS buffer and 200 μL of suspension and transferred to microtiter plate (see Supplementary Methods, β-galactosidase assay).

To quantify the signal molecule SM dependent response (RM) at the single-cell level during growth as a function of time and compare this response to signal production, the fluorescently tagged *B. subtilis* PS-216 (*srfA-cfp, comQ-yfp*), which expresses two different fluorescent proteins from two promoters was used. The expression level of *srfA-cfp* is a measure for single cell RM production, while *comQ-yfp* is a measure for single-cell SM production (*comQ* and *comX* share the same promoter and their genetic sequences often overlap[15], therefore expression level of *comQ* corresponds to the expression level of *comX*). The gene expression and co-localization was evaluated by single-cell fluorescence microscopy (see, Gene expression evaluation by single cell fluorescence microscopy, Co-localization analysis).

**Gene expression evaluation by single cell fluorescence microscopy.** The gene expression evaluation of *B. subtilis* cells tagged with yfp and cfp was based on the previously set protocol[61]. Briefly, diagnostic slides (10 wells/6 mm) were coated by transferring ~10 μl of 0.05% (w/v) poly L-lysine into each well and left for an hour at room temperature until water evaporated. The samples of approximately 1 mL of the culture samples of fluorescently tagged and non-tagged *B. subtilis* PS-216 strains were collected at selected time intervals during the incubation in batch fermenter system. After washing the cells by SS buffer, ~15 μl of cell suspension was transferred to each of the coated wells (5 for tagged and 5 wells for non-tagged strains) of a diagnostic slide and incubated for 15 min at room temperature to allow cells to adhere to the wells. Afterwards, the slide was rinsed with physiological solution (0.9% NaCl) to remove the unattached cells and excess fluid was dispersed. To reduce photo-bleaching 2 μl of the SlowFade Gold antifade reagent (Thermo Fisher Scientific, Inc., USA) were added to each well and immediately taken for observation under Axio Observer Z1 epifluorescence microscope (Zeiss, Göttingen, Germany). Differential interference contrast (DIC) and fluorescence images were observed (objective ×100, NA 1.4, Zeiss) and recorded with a coupled MRm Axiocam camera (Zeiss) operating in 2 × 2 binning mode. The filter-sets used for fluorescent contrast were CFP 47 HE and eYFP 46 HE (Zeiss). For most of the time points at least 1000 bacteria were examined for their fluorescence.

For quantification and flat field correction the fluorescence sample microscopy images were firstly normalized on the sodium fluorescein standard. To take into account also potentially non-fluorescent cells, the DIC images of the same field of view were taken simultaneously with the fluorescent images. The normalized fluorescence intensity of single cells is then extracted by combining the location information about all cells in the DIC images with the fluorescence images via custom made ImageJ script (for the script refer to previously set protocol[61]). In order to take into account the autofluorescence of the cells, the same procedure was applied to the images of control samples with the non-fluorescently tagged cells. Then, by using OriginPro software (OriginLab, Massachusetts, USA) we were able to obtain the lognormal distribution of fluorescent marker (i.e., Yfp or Cfp). The procedure is however not trivial, as the fluorescence intensity of tagged cells contains besides the fluorescence intensity of the marker also autofluorescence that cannot be simply subtracted. Instead, one can assume the two fluorescence intensity contributions are independent and in this case the distribution of fluorescence intensity of the tagged cells is the convolution of distribution of the

fluorescence intensity of the marker (i.e., Yfp or Cfp) and the distribution of background fluorescence intensity (autofluorescence). To extract the parameters of unknown distribution of fluorescent marker, we fitted the distribution of fluorescence intensity of tagged cells by convolution of background fluorescence intensity (autofluorescence) and unknown lognormal fluorescent marker distribution. For details refer to our previous study[61].

The outliers in Yfp or Cfp distributions (i.e., hyperproducers) were determined using Grubb's statistical test in OriginPro software, which is based on two-sided student $t$-test, by setting the significance level to 0.05.

**Co-localization analysis**. To spatially match fluorescence intensity of *srfA-cfp* with *comQ-yfp* simultaneously expressed in the same cells the custom script was written and run in OriginC[62]. The same procedure was applied also to the control, i.e., the cells without fluorescence markers (autofluorescence). From the two datasets two correlation graphs were plotted (OriginPro) and linear Pearson correlation coefficient determined. As the correlation coefficient of the labelled cells also encompasses the correlation of autofluorescence, one cannot directly evaluate the correlation of Cfp with Yfp. Therefore, one has to compare the correlation coefficient of the control cells with fluorescently labeled cells. To further aid the interpretation of the correlation we simulated the intensities of labeled cells with an assumption that no correlation in Cfp and Yfp is present. The fluorescence intensities calculated on the basis of lognormal distributions obtained from the gene expression evaluation with single cell microscopy (see Gene expression evaluation with single cell fluorescence microscopy) were randomly added to the fluorescence intensities control cells for each fluorescence channel independently. The correlation coefficient of this set of data that represented the no correlation case of Cfp with Yfp was then used for comparison with the correlation coefficients of labeled and control cells.

**Data modeling and simulation of the response to the signal molecule and signal molecule production**. The analytical form of the model equation for response per cell to the signal during bacterial growth in the fermenter batch system (Eq. 3, eq S11) was derived and fitted to the experimental data by the help of Wolfram Mathematica 11.0 (Wolfram Research, Inc.). Simulation of various modes of normalized response as a function of bacterial density was performed in OriginPro software (OriginLab, Massachusetts, USA). The fitting of experimental data of signal molecule (SM) concentration as a function of the bacterial density and normalized response, *W(SM)*, as a function of SM concentration was performed using allometric function (Eq. 1) and Hill equation (Eq. 2), respectively. The fitting by logistic equation of growth curves (eq S12) was performed in OriginPro as well. To minimize reduced $\chi^2$ that was used as a criterion for goodness of fit, the Levenberg–Marquardt method was applied (OriginPro). The lowest level of response (LLR) was determined from the graph of the normalized response as a function of SM (ComX) concentration as the first significantly positive value of the response (i.e., the lower 95% confidence level > 0) in the direction of increasing ComX concentration.

**Statistics and Reproducibility**. Statistical analysis was performed using the OriginPro (OriginLab, Massachusetts, USA) or Wolfram Mathematica 11.0 (Wolfram Research, Inc.). Unpaired Student's $T$-test, two-sided, was used to calculate the statistical significance of data sets. A $P$ value of less than 0.05 was considered statistically significant. The model fits are shown together with 95% confidence level. The measure of the quality of fits was reduced $\chi^2$ and $R^2$. The error of obtained values of fitting parameters is the standard error (SE), unless stated otherwise. To determine the number of hyper-expressing cells, Grubb's test was used. If experiments were performed in microtiter plates, at least 8 wells were used as replicates within a biological replicate. In experiments with larger volumes at least two Erlenmeyer flasks were used within an experiment. On top of that, we always performed all our experiments in at least three biologically independent experiments. In microplate assays, we performed statistical analysis by using modified Z-score to exclude the outliers. The lower limit of detection (LOD) for biosensor assay for ComX concentration measurements was calculated from calibration curves and was defined as the minimal ComX concentration that induces the response in the biosensor BD2876 that significantly differs from the response of the blank. Therefore, the LOD was determined as ComX concentration at the response that equals to the response of the blank + 3 standard deviations of the response of the blank.

**Reporting summary**. Further information on research design is available in the Nature Research Reporting Summary linked to this article.

## Data availability

The authors declare that the relevant data supporting the findings of this study are available in the article and its Supplementary Information files, or from the corresponding author upon request.

## Code availability

The ImageJ macro code[60] for extracting the fluorescence intensities of *B. subtilis* PS-216 (Δ*comQ* P*srfAA*-*yfp*) and the OriginC code[62] for matching the fluorescence intensities of

Cfp and Yfp in *B. subtilis* wt PS-216 (*comQ-yfp*, *srfA-cfp*) imaged by fluorescence microscopy are freely available on Github by https://doi.org/10.5281/zenodo.4205585 and https://doi.org/10.5281/zenodo.4206230, respectively.

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

## Acknowledgements
We thank Vesna Vogrič and Martina Vrhar for their contribution with initial protocol development for ComX measurements and isolation. This work has been supported by the Slovenian Research Agency under the grant J4-9302, P4-0116 and the University infrastructural center "Microscopy of biological samples" located in Biotechnical faculty, University of Ljubljana.

## Author contributions
I.D. and I.M. conceived the project with the input from other authors. Modeling and most data analysis was performed by I.D. Experiments were conducted by M.S., A.D., and Ž.P., by supervision of I.D., T.D., and I.M. I.M. was responsible for funds and project management. The manuscript draft was written by I.D. and improved by I.M., A.D., and other authors.

## Competing interests
The authors declare no competing interests.
