## [Peer Review File · Communications Biology]

Reviewers' comments:

Reviewer #1 (Remarks to the Author):

Summary

The goal of the paper is to see whether the *Bacillus subtilis* comQXPA quorum-sensing system represents a "true quorum-sensing system" – a communication system that shows a sharp transition from minimal to maximal response. In gram-negative bacteria, this is achieved by a feedback-loop where the signal is positively regulated by the response it activates, causing a sharp transition in signal production and response as a function of density. The comQXPA system does not follow this paradigm, as signal production is decoupled from the response. The authors show that the external signal concentration has a quadratic dependence on cell density, and that due to the high specificity of the system, signal production carries no quantifiable fitness cost. Surprisingly, the authors show that oxygen concentration is the environmental cue regulating signal production. The authors also show that the response is hypersensitive to the signal, and that both response and signalling gene expression are highly variable in the population. Together, the authors show that the comQXPA system does indeed display the features the authors claim to constitute a "true quorum-sensing" system.

Major comments

I think the authors provide a solid set of experiments to characterize the comQXPA quorum-sensing system. I was especially struck by the result of the effect of oxygen availability on signal production.

My main concern is about the style of the figures, which I think can be significantly improved. This is not a critical, but a stylistic issue.

From an experimental viewpoint, I think the authors can improve their claim of a quadratic dependence between signal concentration and population density by displaying the results on a log-log scale, and providing a more thorough analysis of their limit of detection of the signal. A possible explanation for the observed pattern is that there is a linear dependence, where the response at lower ODs is not accurately characterized due to limit of detection issues.

Stylistic/technical issues

This paper is plagued by the use of two or more different scales on the same axis. This does not help the authors deliver the message of the underlying data, and indeed inhibits it. Most plots should be broken up to avoid this problem. Additionally, the authors repeatedly display results from a single experiment, characterizing the much less informative technical variation in the error bars. This should be changed to display the actual day-to-day variation in the results (i.e., the biological variation). The meaning of the different symbols is not mentioned in the caption of figure 4.

Reviewer #2 (Remarks to the Author):

This is an interesting description of peptide signaling in *B. subtilis* and how it acts as a "true" QS system despite lacking an autoinduction mechanism.

The main issue that I have is with the data that begin in the section that starts on line 216. I had a difficult time ascertaining, even after reading the materials and methods, exactly what it is that the authors are measuring here. My primary concern is that the range of concentrations tested includes an inflection point, below which there is < 1 signal molecule per cell and above which there is > 1 . This depends on the cell density, which I could not determine. If there is such an inflection point, it is the natural explanation for the observed "non-linear" response seen.

Other comments for the authors consideration:

1. The introduction is a bit rambling with a long discussion of srfA regulation (lines 77-84) which is not really germane to the paper.

2. At least in on system, the AHL QS autoinduction system has been "broken" and still resulted in QS activation, although it was non-synchronous (Scholz and Greenberg, 2017) -- it is probably worth adding this to the discussion.

3. Figure 2a - different colors should be chosen. It's really hard to distinguish between dark purple and black!

4. Line 184 - "correlation" might be more appropriate than "agreement"

5. Figure 3 - the multiple-axis graphs are basically indecipherable (double left y-axis and single right y-axis).

Point by point reply to the reviewers' comments:

We prepared our responses in blue ink. Modified figures, including figure legends, are given at the end of this document, as requested by the guidelines of the journal. The changes in the main text file and the supplementary are marked with yellow.

Reviewer #1 (Remarks to the Author):

Summary

The goal of the paper is to see whether the *Bacillus subtilis* comQXPA quorum-sensing system represents a “true quorum-sensing system” – a communication system that shows a sharp transition from minimal to maximal response. In gram-negative bacteria, this is achieved by a feedback-loop where the signal is positively regulated by the response it activates, causing a sharp transition in signal production and response as a function of density. The comQXPA system does not follow this paradigm, as signal production is decoupled from the response. The author’s show that the external signal concentration has a quadratic dependence on cell density, and that due to the high specificity of the system, signal production carries no quantifiable fitness cost. Surprisingly, the authors show that oxygen concentration is the environmental cue regulating signal production. The author’s also show that the response is hypersensitive to the signal, and that both response and signalling gene expression are highly variable in the population. Together, the authors show that the comQXPA system does indeed display the features the authors claim to constitute a “true quorum-sensing” system.

We are happy to read such a well-written summary of our work because it means that we were able to pass our message to the reviewer and hopefully to the future readers. We are also grateful to the reviewer for carefully reading our work and for constructive comments.

Major comments

I think the authors provide a solid set of experiments to characterize the comQXPA quorum-sensing system. I was especially struck by the result of the effect of oxygen availability on signal production. My main concern is about the style of the figures, which I think can be significantly improved. This is not a critical, but a stylistic issue.

From an experimental viewpoint, I think the authors can improve their claim of a quadratic dependence between signal concentration and population density by displaying the results on a log-log scale, and providing a more thorough analysis of their limit of detection of the signal. A possible explanation for the observed pattern is that there is a linear dependence, where the response at lower ODs is not accurately characterized due to limit of detection issues.

We made the limit of detection (LOD) analysis of our method for signal molecule (ComX) determination. The LOD, which we defined as the minimal ComX concentration that induces the response in the biosensor BD2876 that significantly differs from the response of the blank. This means that we determined the ComX concentration at the response that equals to the response of the blank

(i.e. response at ComX = 0 nM) + 3 standard deviations of the response of the blank. Although each time point has its own calibration curve and thus its own LOD, the calculated LOD was in general at 0.1 nM ComX concentration, indicating that most of our measurements points were above this limit. The LOD is marked now as a dotted line in Fig. 2b. This figure is now displayed as suggested by the reviewer, as a log-log plot. The best fit to the points (\geq LOD), blue line, has a slope of 2 (i.e. quadratic dependence), whereas the best fit of a line (black) with the slope 1 (i.e. linear dependence) does not fit the data. One can now clearly see that there is a quadratic dependence between signal concentration (ComX) and population density. By fitting only points \geq LOD, the best-fit parameters changed only marginally; slope a is now 2.09 ± 0.10 , and before, when all points were taken into the fitting procedure, slope a was 2.05 ± 0.10 ; parameter b stayed the same within 2 significant digits. We changed the values in the text accordingly (lines 131, 135).

We have also modified the Fig. 2b accordingly and its figure caption. We also added the information about the calculation of LOD to the "Statistical analysis" chapter in the manuscript:

Lines 515-520: "The lower limit of detection (LOD) for biosensor assay for ComX concentration measurements was calculated from calibration curves and was defined as the minimal ComX concentration that induces the response in the biosensor BD2876 that significantly differs from the response of the blank. Therefore, the LOD was determined as ComX concentration at the response that equals to the response of the blank + 3 standard deviations of the response of the blank."

Stylistic/technical

issues

This paper is plagued by the use of two or more different scales on the same axis. This does not help the authors deliver the message of the underlying data, and indeed inhibits it. Most plots should be broken up to avoid this problem. Additionally, the authors repeatedly display results from a single experiment, characterizing the much less informative technical variation in the error bars. This should be changed to display the actual day-to-day variation in the results (i.e., the biological variation). The meaning of the different symbols is not mentioned in the caption of figure 4.

When one wants to include the information on how the quantities that are dependent on the same variable correlate, the most direct way is to use secondary y-axis. However, we agree with the reviewer that more different scales on the same axis, especially if there are more than two scales, makes the reading of the graphs more difficult.

Therefore we decomposed the each of the two complex graphs in Fig. 3 that had on y-axis dissolved oxygen (DO), β -galactosidase activity and optical density of the culture (OD650), to two graphs: one with DO on the primary y-axis and OD650 on the secondary y-axis and the second graph that has only β -galactosidase activity on y-axis. Graphs are now indeed easier to read.

We also modified the figure caption accordingly.

Fig. 4b also has 3-quantities on the same graph. We decided for this type of presentation because our communication model directly relates the Response per cell ($RM(t)/N(t)$) with the growth (Optical density, OD650), and the signal molecule concentration (SM- ComX) and the correlation among these quantities can be in the same graph directly observed. Also, we wanted to show, where is the lower limit of response (LLR, the smallest quantity of ComX that induces a measurable response in PS-216 strain) determined from Fig. 4a and what is OD650 and the Response at that time. However, to simplify

Fig. 4b, we removed the third-y scale, as the information about the absolute amounts of ComX is not crucial to understand the figure. We also added the missing symbols of Fig. 4a in its caption.

Also in Fig. 2a we have, in addition to the primary y-axis (OD650, growth of the ComX producing strain), included the secondary y-axis (ComX concentration), but we believe that in this case, the correlation between growth and ComX production has to be plotted on the same graph because we wanted to relate the two quantities.

Considering our presentation of single experiments we would first like to explain why we decided for this kind of presentation. We have chosen to do so, because the biological replicates were often not measured at precisely the same times during growth. Therefore, the data cannot be simply averaged and then SD calculated. Also, often the two quantities presented on the same graph are dependent through the same experiment and by averaging several experiments, the information about the correlation of the two quantities might be lost. Alternatively, we could display several biological replicates on the same graph, but in some cases the graphs can then become unclear. Therefore, to address this comment we have considered each case, where we have shown only one biological replicate, separately, and made the changes to show biological variability.

In Fig. 2a we added the data of two biological replicates and marked them with different symbols. We used the same approach also in Fig. 2b. In Fig. 2c, the displayed curves are already the averages of three replicates and also the calculated growth constants are based on 3 biological replicates, which was before not clearly stated.

To show biological variability for Fig. 3 we upgraded the Fig. S4 to show in addition to positive control (isolated ComX diluted in Δ ComQ spent medium) also three independent experiments of negative control (Δ ComQ spent medium) and experimental (10x diluted wt spent medium in Δ ComQ spent medium) in conditions with supplied and limited oxygen.

The growth curves of strain PS-216 (srfA-lacZ) are fitted by logistic equation and response curves during growth by our model equation. One of these curves is shown in Fig. 4b. To show the variability in growth curves, the average fitting parameters of three replicates including standard deviation is given now in additional Table S3. As can be seen, the parameters have relatively small variation indicating biological variability in the growth of PS-216 (srfA-lacZ) was low. To show the variability in response curve, we upgraded Table S4 to show the average and standard deviation of 5 replicates. Also in this case it can be seen that the curves describing response data displayed only limited variation.

We added references to the tables, updated the values of the parameters in the main text and also included a few lines of the text about logistic equation that were missing before into Supplementary:

Lines 868-875:” Note that for the purpose of fitting the $N(t)$ was expressed as the optical density of bacterial culture (OD650). As our fitting procedure by eq 3 required a continuous function of OD650, the measured OD650 was fitted by logistic equation:

$$OD_{650}(t) = A2 + \frac{(A1-A2)}{1+\left(\frac{t}{t0}\right)^p} \quad \text{eq S13}$$

where t is time of bacterial growth, $A1$ the lower horizontal asymptote and $A2$ the upper horizontal asymptote, $t0$ is the time denoting midpoint of growth and p describes the steepness of the growth curve.”

To give the idea of variability in distributions of mean fluorescence intensity per cells expressing comQ-yfp (Fig. 5g) and srfA-cfp (Fig. 5h), we added another set of distributions in Supplementary as Fig. S8 and added references to it in the main text.

Reviewer #2 (Remarks to the Author):

This is an interesting description of peptide signaling in *B. subtilis* and how it acts as a "true" QS system despite lacking an autoinduction mechanism.

We thank the reviewer for reviewing our work, giving constructive comments and showing interest in our results.

The main issue that I have is with the data that begin in the section that starts on line 216. I had a difficult time ascertaining, even after reading the materials and methods, exactly what it is that the authors are measuring here. My primary concern is that the range of concentrations tested includes an inflection point, below which there is < 1 signal molecule per cell and above which there is > 1 . This depends on the cell density, which I could not determine. If there is such an inflection point, it is the natural explanation for the observed "non-linear" response seen.

To make this part clearer and to better explain what is measured, we modified the first paragraph, including the title:

Lines 216-226: **"The response model shows that the response of the cells to ComX is non-linear**

In our model, the expression level of the *srfA* operon serves as a measure for the response (RM) to the signal molecule SM, represented by ComX. To study how RM depends on SM we evaluated promoter activity of *srfA* in the *B. subtilis* PS-216 (Δ comQ, *PsrfAA-yfp*), which carries the markerless deletion of comQ42 and is therefore signal-deficient. Response level was assessed by incubating the PS-216 (Δ comQ, *PsrfAA-yfp*) for 4 hours in the presence of different ComX concentrations, which was the only factor that varied in this experiment. The response level was expressed as yfp fluorescence per cell, normalized to the maximum response, W_{max} , which gives a relative measure, $W(SM)$, of how strongly the cells respond to ComX and this is shown as a function of the exogenously added ComX in Fig. 4a."

We agree with the reviewer that if there was < 1 signal molecule (SM) per cell, then this might be the reason for the inflection point in the Response curve. However, this is not the case because the ratio SM/cell was $\gg 1$, which can be calculated by:

1. Typical OD650 was after 4 h of incubation (as in Fig. 4a) around 1 a.u. corresponding to 4×10^8 cells/mL
2. The lowest concentration of SM (ComX) tested was $0.17 \text{ nM} = 0.17 \times 10^{-9} \text{ mol/L} = 0.17 \times 10^{-12} \text{ mol/mL}$

This means that there was 0.17×10^{-12} mol of SM per 4×10^8 cells, which gives 4.3×10^{-22} mol of SM per cell.

As 1 mol is 6×10^{23} SM this gives 260 SM/cell; by the time of inoculation (2 % inoculum), the ratio was even higher, about 10 000 SM/cell.

Therefore, during this experiment, the ratio of SM/cell was at least two orders of magnitude higher than 1 SM/cell at all time points and tested concentrations.

Other comments for the authors consideration:

1. The introduction is a bit rambling with a long discussion of *srfA* regulation (lines 77-84) which is not really germane to the paper.

We agree that the description was too long. We have shortened it to one sentence:

Lines 78-81: "Although the *srfA* expression required for the synthesis of the major lipopeptide antibiotic surfactin²⁰ also depends on other extracellular peptide signaling systems from the Rap-Phr family^{21,22}, the research of this paper is focused on ComX dependent response."

2. At least in on system, the AHL QS autoinduction system has been "broken" and still resulted in QS activation, although it was non-synchronous (Scholz and Greenberg, 2017) -- it is probably worth adding this to the discussion.

Thank you for bringing our attention to this interesting paper. We have cited it in the Introduction section, and changed the references accordingly.

3. Figure 2a - different colors should be chosen. It's really hard to distinguish between dark purple and black!

We have replaced dark purple with green in Fig. 2. In Fig. 2 green color is now reserved for signal molecule (SM), whereas black and red are for bacterial strains.

4. Line 184 - "correlation" might be more appropriate than "agreement"

We have replaced the "agreement" with "correlation"

5. Figure 3 - the multiple-axis graphs are basically indecipherable (double left y-axis and single right y-axis).

To simplify the presentation, we decomposed each of the two complex graphs in Fig. 3 that had on y-axis dissolved oxygen (DO), β -galactosidase activity and optical density of culture (OD650), to two graphs, one with DO on primary y-axis and OD650 on secondary y-axis and the second graph that has only β -galactosidase activity on y-axis. Graphs are now indeed easier to read.

Figure 2. The accumulation of signal molecule (SM) during the growth of *B. subtilis* (a, b) and fitness cost of SM production (c, d). In (a): the growth curve (■, ●, ▲) of *B. subtilis* PS-216 $\Delta comP$ (no signal receptor) producing SM (ComX) (■, ●, ▲) that is accumulating in the growth medium of fermenter working in the batch mode; the shape of the symbol denotes the three biological replicates. The error bars for OD650 represent the uncertainty of the OD650 measurements; the error bars for SM concentration are standard errors calculated from 8 technical replicates. In (b): best fit (—) and corresponding 95 % confidence level (- - -) of the power-law model (eq 1) to the experimental data (■, ●, ▲) of three biological replicates where the data \geq limit of detection of SM (···); the error bars for SM concentration are calculated from 8 technical replicates for each biological replicate; best fit (—) of the linear function; In (c):

the growth curves of *B. subtilis* PS-216 (c) with no signal molecule receptor ($\Delta comP$) (■) and no signal molecule production and receptor $\Delta comQXP$ (■) represent an average of three biological replicates. The OD650 at $t = 0$ h was corrected with respect to the measured CFU of the inoculum. The slopes of the fitted lines in (c) correspond to the growth rate divided by \log_2 ; the exponential phase points in the most reliable OD650 region (> 0.1 a.u. and < 0.7 a.u.) were considered. The slopes do not differ significantly ($P = 0.32$): $\Delta comP = (0.496 \pm 0.007) \text{ h}^{-1}$ and $\Delta comQXP = (0.503 \pm 0.008) \text{ h}^{-1}$. In (d): the same strains grown in co-culture; each time OD650 reached 0.6 a.u. the co-culture was transferred to the fresh medium. The error bars represent SD of three biological replicates (c, d). The green line crosses all error bars indicating that there are no significant differences (d).

Figure 3. Influence of O_2 on the presence of SM (ComX). The strain PS-216 wt was grown in the fermenter batch system where oxygen supply was limited (a, c) or supplied to the saturation (b, d). In (a,b): The growth was monitored by OD650 (■) and oxygen saturation was followed by a polarizable electrode (□). In (c, d): the samples of spent medium were periodically taken to test the presence of ComX via β -galactosidase activity of the ComX biosensor BD2876. The biosensor was incubated in the fresh CM medium supplemented with either spent medium of the $\Delta comQ$ strain (no ComX, negative control) ▲, or supplemented with the wt strain spent medium 10 times diluted by spent medium of the $\Delta comQ$ strain (experimental) ▲; the spent medium of the $\Delta comQ$ strain was obtained in the parallel batch system; one experimental replicate is presented with error bars representing SD of 8 technical replicates; additional replicates are presented in Fig. S4

Figure 4. Maximum normalized response per cell (a) of the strain *B. subtilis* PS-216 ($\Delta comQ$, P_{srfA} -*yfp*) (no signal production) to the exogenously added signal molecule, SM, (ComX) and response per cell of *B. subtilis* PS-216 (*srfA-lacZ*) (signal producer and responder) during the growth in batch mode (b). In (a): *B. subtilis* PS-216 ($\Delta comQ$, P_{srfAA} -*yfp*) was incubated in the presence of SM for 4 hours and response was measured as the activity of the *srfA* promoter. Four independent experiments were performed (\blacktriangle , \bullet , \blacksquare , \blacklozenge). Best, concatenated fit (—) to the model in eq 2 is presented together with 95 % confidence level (---). In (b): the logistic fit (—) to the growth curve, measured as OD650 (\blacksquare) of the culture *B. subtilis* PS 216 (*srfA-lacZ*) producing signal molecule, SM, (----) that accumulated in the growth medium of batch system. The response per cell, measured as the β -galactosidase activity of *srfA* promoter of *B. subtilis* PS-216 (*srfA-lacZ*) (\blacksquare) was fitted by eq 3, ($R^2 > 0.99$). The time interval at which SM concentration is high enough to cause the measurable response, i.e. lower limit of response (LLR) as predicted from data in experiments in (a) is given as dashed window in (b). One of the five qualitatively and quantitatively similar experimental results is presented (b). Error bars represent SD of 8 technical replicates. For fits of additional replicates and data variability refer to Table S3 and S4.

Figure S4: β -galactosidase activity of the ComX biosensor BD2876 that was incubated in the fresh CM medium supplemented with either spent medium of the $\Delta comQ$ strain (no ComX, negative control) $\blacktriangle, \blacksquare, \bullet$, or supplemented with spent medium of the PS-216 ($\Delta comQ$) strain with added isolated ComX (≈ 1 nM, positive control) $\blacktriangle, \blacksquare, \bullet$, or supplemented with the wt strain spent medium 10 times diluted by spent medium of the $\Delta comQ$ strain (experimental) $\blacktriangle, \blacksquare, \bullet$; The spent medium of the PS-216 ($\Delta comQ$) strain was obtained in the parallel fermenter batch system. The strains PS-216 wt and PS-216 ($\Delta comQ$) were grown in the fermenter batch system where oxygen supply was limited (a) or supplied to the saturation (b). The shape of the symbol corresponds to 3 independent experiments. The ComX biosensor BD2876 barely responded to the spent medium with no ComX (PS-216 $\Delta comQ$ spent medium) and strongly responded to the same medium when purified ComX was added, indicating that ComX is the major factor controlling *srfA-lacZ* expression in biosensor BD2876. However, the same quantity of ComX added to the spent media collected at different time points does not induce the same level of response by biosensor, therefore for quantification of ComX (Fig 2a,b) the $\Delta comQ$ spent medium used for construction of calibration curve was collected for each time point separately (see Material and methods, Biosensor based quantification of ComX concentration).

Figure S8. The second replicate (for first replicate, see Fig 5g,h) of the distribution of gene expression measured by single cell fluorescence microscopy expressed as Na-fluoresceinate standard normalized mean fluorescence intensity per cells expressing *comQ-yfp* (a) and *srfA-cfp* (b). ON is the overnight culture; areas under the curves are the same at all time points.

parameter	value	Standard deviation
A1	0.029	0.004
A2	0.75	0.02
t0	2.61	0.05
p	4.8	0.5

Table S3. The average values of fitting parameters values of fit of 3 growth curves of PS-216 (*srfA-lacZ*) by eq S13 (see Fig. 4b). The R^2 of all fits was ≥ 0.99 and reduced $\chi^2 < 3 \times 10^{-4}$. Relatively small standard deviation indicates the growth curves were very similar.

parameter	value	Standard deviation	t-statistic	P-value
k	760	120	>15	$<1 \times 10^{-7}$
RMO	5.5	1.5	>9	$<7 \times 10^{-6}$

Table S4. The average values of fitting parameters values and statistics of fits of 5 experimental sets by eq 3 (see Fig. 4b), which represents the comQXPA communication system model. T-test was two-sided. The R^2 of all fits was ≥ 0.95 . Relatively small standard deviation indicates the response curves were very similar.

REVIEWERS' COMMENTS:

Reviewer #1 (Remarks to the Author):

The authors have addressed all my comments fully to my satisfaction.

Reviewer #2 (Remarks to the Author):

The authors have addressed my concerns with the prior manuscript.